



# Limited angle tomography of mesoscale gravity waves by the infrared limb-sounder GLORIA

Isabell Krisch[1], Jörn Ungermann[1], Peter Preusse[1], Erik Kretschmer[2], and Martin Riese[1]

[1]Forschungszentrum Jülich, Institute of Energy- and Climate Research, Stratosphere (IEK-7), Jülich, Germany
[2]Karlsruhe Institute of Technology, Institute of Meteorology and Climate Research, Karlsruhe, Germany

**Correspondence:** Isabell Krisch (i.krisch@fz-juelich.de)

**Abstract.** Three-dimensional measurements of gravity waves are required, in order to quantify their direction resolved momentum fluxes and get a better understanding of their propagation characteristics. Such 3-D measurements of gravity waves in the lowermost stratosphere can be provided by the airborne Gimballed Limb Observer for Radiance Imaging of the Atmosphere (GLORIA) using full angle tomography. Closed flight patterns of sufficient size are needed to acquire the full set of angular
measurements for full angle tomography. These take about two hours and are not feasible everywhere due to scientific reasons or air traffic control restrictions. Hence, this paper investigates the usability of limited angle tomography for gravity wave research. Limited angle tomography uses only a limited set of angles for tomographic reconstruction and can be applied on linear flight patterns. A synthetic end-to-end simulation has been performed to investigate the sensitivity of limited angle tomography towards gravity waves with different wavelengths and orientations in respect to the flight path. For waves with wave fronts
roughly perpendicular to the flight path, limited angle tomography and full angle tomography can derive wave parameters like wavelength, amplitude and wave orientation with similar accuracy. For waves with horizontal wavelength above $200\,\mathrm{km}$ and vertical wavelength above $3\,\mathrm{km}$, the wavelengths can be retrieved with less than $10\%$ error, the amplitude with less than $20\%$ error and the horizontal wave direction with an error below $10°$. This is confirmed by a comparison of results obtained from full angle tomography and limited angle tomography for a real measurement case on 25 January 2016 over Iceland. The
reproduction quality of gravity wave parameters with limited angle tomography, however, depends strongly on the orientation of the waves with respect to the flight path. Thus, full angle tomography might be preferable in cases where the orientation of the wave cannot be predicted or waves with different orientations exist in the same volume and thus the flight path cannot be adjusted accordingly. Also for low amplitude waves and short scale waves full angle tomography has advantages due to its slightly higher resolution and accuracy.

## 1   Introduction

Gravity waves (GWs) couple the atmosphere vertically by transporting energy and momentum from the surface to altitudes as far as the mesosphere. On this way through the atmosphere, GWs interact with the mean flow and, thus, are responsible for the wind reversal in the mesosphere and influence prominent circulation patterns such as the quasi-biennial oscillation (QBO) of stratospheric tropical winds, and the meridional Brewer-Dobson circulation (BDC) in the stratosphere. These circulation



patterns can then affect surface temperature and pressure patterns (Sigmond and Scinocca, 2010; Kidston et al., 2015; Sandu et al., 2016; Scaife et al., 2016).

Due to their small scales, GWs are implemented into many climate projection and weather prediction models in the form of simplified sub-models, so-called GW parameterisations. For practical reasons, these parameterisations assume solely ver-
tical propagation of GWs. However, multiple studies highlight the importance of 3-D propagation (Preusse et al., 2009; Sato et al., 2009; McLandress et al., 2012; Kalisch et al., 2014; Ribstein and Achatz, 2016). To gain a better understanding of 3-D propagation and improve the GW parameterisations in models, 3-D measurements of GWs are required (Geller et al., 2013).

The Gimballed Limb Observer for Radiance Imaging of the Atmosphere (GLORIA) can provide such 3-D measurements of gravity waves (Krisch et al., 2017). GLORIA is an airborne infrared limb sounder, which can change its horizontal viewing
direction from $45°$ (right forward) to $135°$ (right backward) with respect to the aircraft's heading (Friedl-Vallon et al., 2014; Riese et al., 2014). If a volume of air is measured from all surrounding angles, it may be reconstructed using tomographic methods (Natterer, 2001). Measuring emitted radiation from all $360°$ around the volume, for instance, by flying a circle, is called full angle tomography (FAT). This technique improves the horizontal resolution of limb sounders by an order of magnitude (Ungermann et al., 2010b). In contrast to FAT, limited angle tomography (LAT) does not measure the volume from
all sides but only from a limited set of angles. Due to the horizontal scanning capabilities of the GLORIA instrument, this is already possible on a linear flight path. However, LAT inversion problems are in general seriously ill-posed (Natterer, 2001).

Since flying in a circular pattern of sufficient size can take more than two hours, FAT is only suitable for measurements in steady atmospheric states, where the conditions do not change during the acquisition time. Accounting for the change of the atmosphere during acquisition is possible for trace gas retrievals by including advection (Ungermann et al., 2011). The
temperature structure, however, is governed by a multitude of waves with different spatial and temporal scales. Our apriori knowledge of the temporal development of these waves is not sufficient to retrieve a fast changing temperature structure using FAT. Furthermore, air traffic control restrictions and trade-offs with other instruments can force linear flight patterns.

Using FAT, a cylindrical 3-D volume with about $400\,\mathrm{km}$ hprizontal diameter and several kilometres in altitude can be reconstructed (Ungermann et al., 2011; Kaufmann et al., 2015; Krisch et al., 2017). In contrast, the 3-D volume, which is
reconstructed using LAT, has a horizontal extent of only $100\,\mathrm{km}-200\,\mathrm{km}$ perpendicular to the flight path depending on the altitude (Ungermann et al., 2011). Is this horizontal extent sufficient to derive the 3-D orientation of GWs? In this paper we will investigate, how well the wave structures are retrieved using LAT and to which accuracy wave parameters can be derived from LAT temperature retrievals.

In order to assess the differences between FAT and LAT using the GLORIA instrument quantitatively, we will derive the
observational filter of both methods. The observational filter is a measure for the sensitivity of an instrument towards measuring different GWs and should be taken into account when comparing measurements from different instruments or measurements with model results (Alexander, 1998; Preusse et al., 2002; Ern et al., 2006; Trinh et al., 2016). In general, an observational filter tells by how much the amplitude or the momentum flux is underestimated by the measurement technique. As it is possible to reconstruct 3-D volumes with GLORIA and, thus, derive a 3-D wave vector, we extend the concept of the observational
filter. Besides the usual observational filter regarding the wave amplitude, we introduce an observational filter for the wave





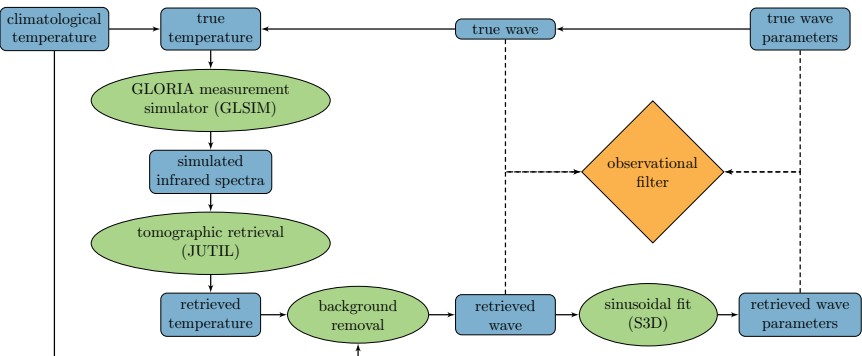

**Figure 1.** Methodological concept of the performed simulation study. A detailed description can be found in Sec. 2.1.

orientation. This observational filter for the wave orientation with respect to the flight direction is important for the planning of research flights: The wave orientation can often be predicted and the flight path adjusted respectively.

This paper is structured as follows: In Sec. 2 a description of the methodological concept on how to derive an observational filter (Sec. 2.1) is followed by a short introduction to the concept of limb sounding and the GLORIA instrument (Sec. 2.2). Afterwards, the methods used for the estimation of the observational filter are described in detail: the GLORIA measurement simulator (Sec. 2.3), the tomographic retrieval concept (Sec. 2.4), the background removal algorithm (Sec. 2.5) and the three-dimensional wave fitting routine (Sec. 2.6). Finally, a definition of the observational filter is given in Sec. 2.7. Section 3 presents and discusses the results of the simulation study for FAT (Sec. 3.1) and LAT (Sec. 3.2) and compares the measurement quality of both methods for a real GLORIA measurement case on 25 January 2016 over Iceland (Sec. 3.3).

## 2 Methods

### 2.1 Methodological concept

The goal of this simulation study is to determine the capability of the GLORIA infrared limb imager to measure mesoscale GWs with limited angle tomography (LAT). The accuracy of reconstructing GW parameters, such as horizontal and vertical wavelength, amplitude and wave orientation is studied. For this purpose, an end-to-end simulation was performed which is described in this section. Figure 1 shows the concept of this end-to-end simulation.

To simulate a realistic atmosphere, a complete climatological field $\boldsymbol{a}_c \in \mathbb{R}^n$ from Remedios et al. (2007) was used. The climatological temperature field $\boldsymbol{T}_c \in \mathbb{R}^m$ was perturbed at each point $\boldsymbol{x}_i \in \mathbb{R}^3$ in space by a simulated, so-called true wave field

$$w_t^i = \hat{T} \cdot \sin\left(\boldsymbol{k}\boldsymbol{x}_i + \phi\right), \tag{1}$$





where $\hat{T}$ is the temperature amplitude, $\boldsymbol{k} \in \mathbb{R}^3$ the 3-D wave vector, and $\phi$ the phase of the wave. In the present simulation study the temperature amplitude $\hat{T}$ is arbitrarily chosen to be 3 K, the wave phase $\phi$ to be $0°$. For an improved conception, from now on, the 3-D wave vector $\boldsymbol{k} = (k_0, k_1, k_2)$ will mainly be expressed in terms of vertical wavelength $\lambda_z = \frac{2\pi}{k_2}$, horizontal wavelength $\lambda_h = \frac{2\pi}{\sqrt{k_0^2 + k_1^2}}$, and horizontal wave direction $\varphi = \arctan2(k_1, k_0)$. Thus, if a downward pointing wave vector

($\lambda_z > 0$, upward propagating wave) is assumed, a wave with wave direction of $0°$ has east-west oriented wavefronts and is tilted towards the north, a wave with wave direction of $180°$ is tilted towards the south. These true waves are added to the climatological temperature field to gain the so-called true temperature field $\boldsymbol{T}_t \in \mathbb{R}^m$.

From this predetermined atmospheric state and with a given flightpath, the GLORIA measurement simulator (Sec. 2.3) calculates a set of infrared spectra, as would be measured by the GLORIA instrument. A tomographic retrieval (Sec. 2.4) is

then performed using these simulated infrared spectra. This retrieval uses only a well-defined set of infrared radiances (Tab. 1) and can reconstruct the atmosphere only in a reduced area, limited by the measurement geometry.

A background removal algorithm then subtracts the climatological temperature field $\boldsymbol{T}_c$ from the retrieved temperature field $\boldsymbol{T}_r \in \mathbb{R}^m$ to obtain the retrieved wave structure $\boldsymbol{w}_r \in \mathbb{R}^m$. In a real measurement case (Sec. 3.3) the background field is unknown and has to be identified by mathematical filtering methods (Sec. 2.5). To solely investigate the sensitivity of the

measurement concept and exclude any additional effects, these filtering methods are not used for the simulation study.

Finally, the retrieved wave structure is compared to the original true wave structure. By repeating this process for different horizontal and vertical wavelengths, the observational filter of LAT is established (Sec. 2.7). To interpret the retrieved wave structure with respect to gravity waves, the wave parameters amplitude, phase and wave vector have to be derived, using the small-volume few-wave decomposition method S3D (Sec. 2.6 and Lehmann et al., 2012). Comparing these retrieved wave

parameters to the prescribed true wave parameters gives detailed information on the usability of the different retrievals for GW research.

## 2.2 The GLORIA infrared limb imager

The Gimballed Limb Observer for Radiance Imaging of the Atmosphere (GLORIA) is an airborne Fourier Transform Spectrometer (FTS) which combines a classical Michelson interferometer with a 2-D detector array Friedl-Vallon et al. (2014).

GLORIA measures the infrared radiation in the spectral range from 780 to $1400\,\mathrm{cm}^{-1}$, which is emitted by molecules in the atmosphere along the line-of-sight (LOS). The interferometer spectrally resolves this radiation to reveal characteristic molecular emissions. A schematic of the limb sounding geometry is depicted in Fig. 2a. Due to the exponentially declining density of the atmosphere with altitude, most radiation along the LOS is emitted at lower altitudes and, thus, around the tangent point. Moreover, for geometrical reasons, a comparatively long part of the LOS samples altitudes close to the tangent point, while

higher atmospheric layers are passed only briefly. As a consequence, conventional limb sounders are more sensitive to changes in the atmosphere around the tangent point (Fig. 2b). The horizontal resolution of conventional limb sounders along the line-of-sight is roughly 200–300 km (von Clarmann et al., 2009; Ungermann et al., 2012). In flight direction the horizontal resolution of 1D retrievals, which mainly depends on the horizontal field of view, is on the order of several kilometres for the airborne limb imager GLORIA.




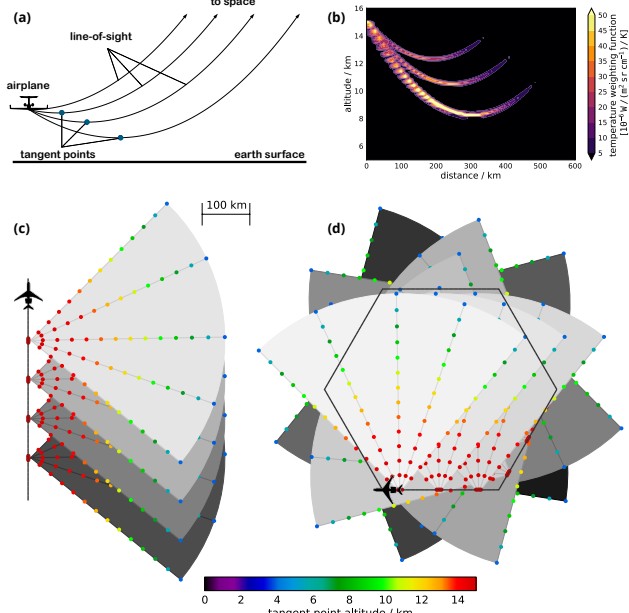

**Figure 2.** In Panel (a) a simple schematic of the limb sounding geometry is given. The lines-of-sight (LOS) have a parabolic shape due to the cartesian coordinate system used for this plot. Panel (b) shows the weighting function along three different LOS indicating the contribution of the respective part of the atmosphere to the observed signal. In panels (c) and (d) the principles of limited angle tomography (LAT) and full angle tomography (FAT) with the GLORIA instrument are depicted, respectively. The dots indicate the tangent points and are coloured according to their altitude. Each grey sector indicates one horizontal scan from $45°$ to $135°$. The lighter the grey, the later in time are these measurements taken.

GLORIA operates in two different modes: The chemistry mode, which has a high spectral sampling of $0.0625\,\mathrm{cm}^{-1}$, and the dynamics mode with a coarser spectral sampling of only $0.625\,\mathrm{cm}^{-1}$. However, the coarser spectral sampling leads to a faster interferogram acquisition and accordingly an improved spatial sampling ($0.45\,\mathrm{km}$ instead of $2.25\,\mathrm{km}$). This improved spatial sampling is used to scan the atmosphere horizontally in steps of $4°$ from $45°$ (right forward) to $135°$ (right backward) with respect to the aircraft's heading. In this way, the same volume of air is measured under different angles, which allows for tomographic retrievals (Sec. 2.4).

Figure 2c and d show the measurement concept for LAT and FAT used in this paper, respectively. In FAT (Fig. 2d), the full volume of the hexagon is covered by measurements from all $360°$ around. In LAT however (Fig. 2c), the air is sampled by measurements from only one side, thus, $90°$ instead of $360°$. A detailed description of the different GLORIA measurement concepts can be found in Riese et al. (2014).





## 2.3 GLORIA measurement simulator

The GLORIA measurement simulator can replicate an infrared spectrum which GLORIA would measure on a flight through a given atmospheric state. In the present paper, for the LAT cases, a linear flight along the zero meridian from 5° S to 5° N was arbitrarily chosen. For the FAT cases, a hexagonal flight pattern with 400 km diameter around 0° N and 0° E is used. The true atmospheric state $\boldsymbol{a}_t \in \mathbb{R}^m$ is composed of a temperature field, a pressure field, and the distributions of several trace gases. The climatological atmospheric state $\boldsymbol{a}_c$ from Remedios et al. (2007) is transformed into the true atmospheric state $\boldsymbol{a}_t$ by inserting the true temperature field $\boldsymbol{T}_t$ (see Sec 2.1). A radiative transfer model $\mathbf{F} : \mathbb{R}^n \rightarrow \mathbb{R}^o$ maps this true atmospheric state onto a set of radiances $\mathbf{F}(\boldsymbol{a}_t)$. As GLORIA can measure radiances only with finite precision, a measurement error $\boldsymbol{\epsilon} \in \mathbb{R}^o$ is added to get the simulated radiances $\boldsymbol{y} \in \mathbb{R}^o$:

$$\boldsymbol{y} = \mathbf{F}(\boldsymbol{a}_t) + \boldsymbol{\epsilon} \tag{2}$$

This straight forward calculation is done with the Juelich Rapid Spectral Simulation Code V2 (JURASSIC2) for each pixel of the GLORIA detector along the chosen flight path leading to a set of thousands of simulated infrared spectra.

This model was developed to efficiently handle imager instruments and tomographic retrievals. It is based on JURASSIC (Hoffmann, 2006), which was previously used as forward model for the evaluation of several satellite- and air-borne remote sensing experiments (e.g. Eckermann et al., 2006; Hoffmann et al., 2008; Weigel et al., 2010; Kalicinsky et al., 2013; Ungermann et al., 2016). It contains several approaches of varying computational complexity and accuracy for computing radiances, but we employ in this work the fast table based approach based on the emissivity growth approximation (EGA; e.g. Weinreb and Neuendorffer, 1973; Gordley and Russell, 1981).

## 2.4 Tomographic temperature retrieval

The JUelich Tomographic Inversion Library (JUTIL) software package is used for mapping the simulated infrared spectra back to the geophysical quantities, in our case the retrieved atmospheric state $\boldsymbol{a}_r \in \mathbb{R}^n$ including the retrieved temperature $\boldsymbol{T}_r$. This retrieval represents an ill-posed problem, which is solved by approximating it with a well-posed one using a Tikhonov regularisation scheme (Tikhonov and Arsenin, 1977). This leads to the minimisation problem:

$$\begin{aligned} J(\boldsymbol{a}) = &\left(\mathbf{F}(\boldsymbol{a}) - \boldsymbol{y}\right)^T \mathbf{S}_\epsilon^{-1}\left(\mathbf{F}(\boldsymbol{a}) - \boldsymbol{y}\right) \\ &+ (\boldsymbol{a} - \boldsymbol{a}_a)^T \mathbf{S}_a^{-1}(\boldsymbol{a} - \boldsymbol{a}_a) \rightarrow \min \end{aligned} \tag{3}$$

with $\mathbf{S}_\epsilon \in \mathbb{R}^{o \times o}$ the measurement error covariance matrix and $\mathbf{S}_a \in \mathbb{R}^{n \times n}$ the covariance matrix of the atmospheric state vector. As an apriori state $\boldsymbol{a}_a$ the climatological field $\boldsymbol{a}_c$ is used. This minimization problem is solved with a truncated conjugate gradient based trust region scheme. More details on the retrieval algorithms used for GLORIA Level 2 processing are described by Ungermann et al. (2015).





Since the temperature perturbations due to the wave are small compared to the background temperature $\boldsymbol{T}_c$, the retrieval can be linearized around this background temperature (Rodgers, 2000; Ungermann et al., 2010a):

$$\boldsymbol{y} - \boldsymbol{y}_a = \mathbf{F}'(\boldsymbol{a}_a)(\boldsymbol{a} - \boldsymbol{a}_a) + \boldsymbol{\epsilon}. \tag{4}$$

$\mathbf{F}'(\boldsymbol{a}_a) = \frac{\partial \mathbf{F}}{\partial \boldsymbol{a}}\big|_{\boldsymbol{a}_a}$ is the Jacobian matrix of the forward model evaluated at $\boldsymbol{a}_a$ and $\boldsymbol{y}_a = F(\boldsymbol{a}_a)$ are the simulated radiances of the background state. With the retrieval gain matrix $\mathbf{G}(\boldsymbol{a}_a) = (\mathbf{F}'(\boldsymbol{a}_a)^T \mathbf{S}_\epsilon^{-1} \mathbf{F}'(\boldsymbol{a}_a) + \mathbf{S}_a^{-1})^{-1} \mathbf{F}'(\boldsymbol{a}_a)^T \mathbf{S}_\epsilon^{-1}$ and the Jacobian matrix $\mathbf{F}'(\boldsymbol{a}_a)$ the averaging kernel matrix $\mathbf{A}(\boldsymbol{a}_a) = \mathbf{G}(\boldsymbol{a}_a)\mathbf{F}'(\boldsymbol{a}_a)$ can be calculated, which converts the true temperature perturbation $\boldsymbol{w}_t = \boldsymbol{a}_t - \boldsymbol{a}_a$ into the retrieved temperature perturbation $\boldsymbol{w}_r = \boldsymbol{a}_r - \boldsymbol{a}_a$:

$$\mathbf{G}(\boldsymbol{a}_a)(\boldsymbol{y} - \boldsymbol{y}_a) = \mathbf{G}(\boldsymbol{a}_a)(\mathbf{F}'(\boldsymbol{a}_a)(\boldsymbol{a} - \boldsymbol{a}_a) + \boldsymbol{\epsilon}) \tag{5}$$

$$\boldsymbol{a}_r - \boldsymbol{a}_a = \mathbf{A}(\boldsymbol{a}_a)(\boldsymbol{a}_t - \boldsymbol{a}_a) + \mathbf{G}(\boldsymbol{a}_a)\boldsymbol{\epsilon} \tag{6}$$

$$\boldsymbol{w}_r = \mathbf{A}(\boldsymbol{a}_a)\boldsymbol{w}_t + \mathbf{G}(\boldsymbol{a}_a)\boldsymbol{\epsilon}. \tag{7}$$

The noise term $\mathbf{G}(\boldsymbol{a}_a)\boldsymbol{\epsilon}$ is negligible with regard to the other terms and is disregarded in the present study. For selected cases, the linear approximation has been validated by a comparison of linear and non-linear retrieval results. The retrievals of the real measurements in Sec. 3.3 are non-linear.

Conventional infrared temperature retrievals for limb sounding instruments are based on optically thin spectral lines for example in the $CO_2$ Q-branch region at $790.75\,\mathrm{cm}^{-1}$ ($12.6\,\mu\mathrm{m}$) (Riese et al., 1997; Ungermann et al., 2010a). Nadir sounders in contrast use spectral lines with different opacity to improve the vertical resolution (Hoffmann and Alexander, 2009). Transferring this concept to limb sounding and including additional lines with high opacity into limb sounding retrievals increases the resolution along LOS (Ungermann et al., 2011). Thus, including some ozone emission lines between $980\,\mathrm{cm}^{-1}$ and $1014\,\mathrm{cm}^{-1}$ with different optical depths in our retrieval and applying LAT improved the horizontal resolution in viewing direction at $10.5\,\mathrm{km}$ altitude to $70\,\mathrm{km}$. The resolution is derived through fitting a 3-D ellipsoid around all points of the averaging kernel matrix $\mathbf{A}$ larger than half the maximum. The horizontal resolution along the flight path is $30\,\mathrm{km}$ and the vertical resolution $400\,\mathrm{m}$. The FAT retrievals have a horizontal resolution of $20\,\mathrm{km}$ in both directions and a vertical resolution of $200\,\mathrm{m}$. The precision of both methods within the tangent point area is below $0.05\,\mathrm{K}$, the accuracy of FAT below $0.5\,\mathrm{K}$, and the accuracy of LAT below $0.7\,\mathrm{K}$.

For the GW sensitivity study a retrieval setup with the spectral ranges 1 to 7 in Tab. 1 is used. For the real measurement retrievals in Sec. 3.3, the spectral ranges 8 and 9 in Tab. 1 are included in the retrieval to improve the knowledge about the $CCl_4$ background radiation in the $CO_2$ Q-branch region. Furthermore, the spectral ranges 10 to 13 are used additionally to retrieve the trace gas $HNO_3$.

To improve the convergence speed and the quality of the real measurement retrievals, apriori fields are taken from different models. The temperature apriori was constructed from European Centre for Medium-Range Weather Forecasts (ECMWF) operational analyses at resolution T1279/L137 by applying the background removal described in Sec. 2.5. The background removal ensures that any GW signature in the retrieval result does not originate from the apriori field. The pressure field was





**Table 1.** Spectral ranges used for the retrievals presented in this paper. The last column indicates the retrieved quantity for each spectral range. For the simulation study (Sec. 3.1 and 3.2), the spectral ranges 1 to 7 are used. For the real measurement retrievals (Sec. 3.3), the spectral ranges 8 to 13 are added.

|  | Spectral range / $\mathrm{cm}^{-1}$ | | | Used for |
|---|---|---|---|---|
| 1 | 790.625 | – | 791.250 | temperature |
| 2 | 791.875 | – | 792.500 | temperature |
| 3 | 956.875 | – | 962.500 | temperature |
| 4 | 980.000 | – | 984.375 | temperature, $O_3$ |
| 5 | 992.500 | – | 997.500 | temperature, $O_3$ |
| 6 | 1000.625 | – | 1006.250 | temperature, $O_3$ |
| 7 | 1010.000 | – | 1014.375 | temperature, $O_3$ |
| 8 | 793.125 | – | 795.000 | $CCl_4$ |
| 9 | 796.875 | – | 799.375 | $CCl_4$ |
| 10 | 883.750 | – | 888.125 | $HNO_3$ |
| 11 | 892.500 | – | 896.250 | $HNO_3$ |
| 12 | 900.000 | – | 903.125 | $HNO_3$ |
| 13 | 918.750 | – | 923.125 | $HNO_3$ |

taken directly from ECMWF. The apriori fields of several trace gases ($CH_4$, $CO_2$, $H_2O$, $O_3$, ...) are taken from the Whole Atmosphere Community Climate Model version 4 (WACCM4).

## 2.5 Background removal

The atmospheric temperature distribution is mainly determined by the stratification and the balanced flow. However, GWs cause small-scale perturbations to this background temperature structure. Before identifying GWs with wave-fitting algorithms, a scale separation of large-scale background and small-scale perturbations has to be performed. This scale separation is called background removal.

For the simulation study in Sec. 3.1 and Sec. 3.2 the background temperature is known to be the climatological temperature field $\boldsymbol{T}_c$. The background removal subtracts this temperature field $\boldsymbol{T}_c$ from the retrieved temperature field $\boldsymbol{T}_r$ and the temperature residual or so-called retrieved wave field $\boldsymbol{w}_r$ remains.

For the retrievals of real measurements presented in Sec. 3.3, one-dimensional Savitzky-Golay filters (Savitzky and Golay, 1964) are applied in all three spatial directions with third-order polynomials over 25 and 60 neighbouring points in vertical and both horizontal directions, respectively. In this way, a spatial separation in large scale background and small scale temperature residuals is achieved.



## 2.6 Three-dimensional sinusoidal fitting routine (S3D)

To compare the retrieval results with the original waves and interpret the structures with regard to gravity waves, wave parameters (horizontal and vertical wavelengths, wave amplitude and wave direction) have to be derived. This is done in overlapping sub-volumes of 5 km vertical and 400 km x 400 km horizontal extent. In these sub-volumes a sinusoid is fitted to the retrieved

wave field $w_r$ using a least square method (Lehmann et al., 2012). In this process, the following equation is minimized:

$$\chi^2 = \sum_i \frac{(f(\boldsymbol{x}_i) - w_r(\boldsymbol{x}_i))^2}{\sigma^2(\boldsymbol{x}_i)} \tag{8}$$

with the sinusoidal function

$$f(\boldsymbol{x}) = \hat{T} \cdot \sin(\boldsymbol{kx} + \phi) = A \cdot \sin(\boldsymbol{kx}) + B \cdot \cos(\boldsymbol{kx}), \tag{9}$$

and a weighting function $\sigma^2(\boldsymbol{x})$, which is chosen to be 1 if a tangent point exists in this grid cell and $10^5$ if not. Systematic

tests of this algorithm by superposition of two sinusoids show that even for wavelengths up to 2.5 times the cube size, the wave parameters of both waves are fitted with errors below 1%.

Due to the fact that the wave structures of the real measurements vary strongly in space, a smaller cube size of 3.6 km x 160 km x 160 km is chosen for the S3D fits presented in Sec. 3.3. This cube size is sufficient to derive vertical wavelengths up to 9 km and horizontal wavelengths up to 400 km.

## 2.7 Observational filter

The relative error of the retrieval within an area A can be calculated as follows:

$$S = \sum_{\boldsymbol{x}_i \in A} \frac{w_t(\boldsymbol{x}_i) - w_r(\boldsymbol{x}_i)}{w_t(\boldsymbol{x}_i)} \tag{10}$$

For a fair comparison, the area A must be chosen in a way, that it covers the measurement region, meaning a region covered with tangent points. In our case, the area A was chosen to be between $9.5 - 11.5$ km altitude, $1.75° - 2.25°$ longitude, and

$-1° - 1°$ latitude for LAT and $9.5 - 11.5$ km altitude, $-1° - 1°$ longitude, and $-1° - 1°$ latitude for FAT.

This relative error is a helpful measure for the reproducibility of GWs by the measurement setup and the retrieval concept. However, it does not give detailed information upon which wave parameters can be derived. Thus, we further define more specific relative errors for the important gravity wave parameters, horizontal $\lambda_h$ and vertical $\lambda_z$ wavelengths, amplitude $\hat{T}$ and horizontal wave orientation $\varphi$:

$$S_\xi = \sum_{\boldsymbol{x}_i \in A} \frac{\xi_t(\boldsymbol{x}_i) - \xi_r(\boldsymbol{x}_i)}{\xi_t(\boldsymbol{x}_i)} \text{ for } \xi \in \lambda_h, \lambda_z, \hat{T}, \varphi \tag{11}$$

These more specific relative errors help to define the quality of our measurement for gravity wave analysis.

The observational filter $O = 1 - S$ is a measure of the sensitivity of an instrument and defines which gravity waves can be detected. The knowledge of the observational filter is necessary for meaningful comparisons of measurements from different instruments or measurement and model results (Alexander, 1998; Preusse et al., 2002; Ern et al., 2006; Ern and Preusse, 2012;

Trinh et al., 2016).





## 3   Results and discussion

### 3.1   Full angle tomography (FAT)

Former studies demonstrated the feasibility of 2-D tomography of rearward looking satellite instruments for the retrieval of 3-D atmospheric structures such as gravity waves (Ungermann et al., 2010a). However, the concept of 3-D tomography with

sidewards looking airborne instruments is quite different and, thus, may exhibit different characteristics. In 2-D tomography a volume is reconstructed from rearward looking measurements on a moving platform which slice the volume into multiple 2-D images. In 3-D FAT, a volume is reconstructed from measurements at different sides all around the volume. Thus, with FAT the problem of wave orientation with respect to instrument and flight path (Ungermann et al., 2010a) does not appear.

Figure 3 shows a 3-D FAT of a GW with horizontal wavelength of 400 km, vertical wavelength of 6 km and horizontal wave

direction $\varphi_t = 180°$. Pictured are three cross sections through the 3-D volume at 10.5 km altitude, at 0° N and at 0° E. The first column shows the true wave, the second column the retrieved wave and the third column the difference of both. Within the tangent point area (dotted lines) the temperature error is below 0.5 K. This is in good agreement with the determined accuracy of 0.5 K (Sec. 2.4).

A S3D fit (Sec. 2.6) was performed for this retrieval at 10.5 km altitude with a cube size of 5 km x 400 km x 400 km. The

results of this fit can be seen in Fig. 3j-m. Within the hexagonal flight pattern the horizontal and vertical wavelength, and the horizontal wave direction are well reproduced. The original amplitude of 3 K is underestimated by 0.1 K.

These S3D results are used to construct the specific observational filters in Fig. 4. A mean value of the S3D fit results between 1° S, 1° N, 1° W, and 1° E gives the specific observational filter of the respective wavelength pair. The horizontal wavelength, the vertical wavelength and the horizontal wave direction are well reproduced for all tested waves. Further, there appears no

phase shift in the FAT retrieval in contrast to conventional 1-D retrievals (Ungermann et al., 2010a). However, the amplitude of the waves is slightly reduced for waves with horizontal wavelength below 200 km or vertical wavelength below 3 km.

This simulation study shows, that FAT is able to properly reconstruct the wave vectors of mesoscale GWs. However, the observational filter of the temperature amplitude has to be taken into account, when comparing these measurements to different data sets.

### 3.2   Limited angle tomography (LAT)

#### 3.2.1   Dependence of the retrieval results on horizontal and vertical wavelengths

Figure 5 shows a comparison of the LAT retrieval results for different wavelengths. The waves in column 1 and 4 have a larger horizontal wavelength of 600 km compared to the waves in columns 2 and 3 with 200 km horizontal wavelength. The vertical wavelength of 6 km of the waves in columns 1 and 3 is longer than the vertical wavelength of 2 km of the waves in columns 2

and 4. The waves with large vertical wavelength in columns 1 and 3 are well reproduced by the LAT retrieval within the tangent point covered area with errors below 0.5 K . The waves with short vertical wavelengths show larger temperature errors of up to 1.5 K within the tangent point area. This difference comes from the curved LOS through the straight wavefronts, which leads





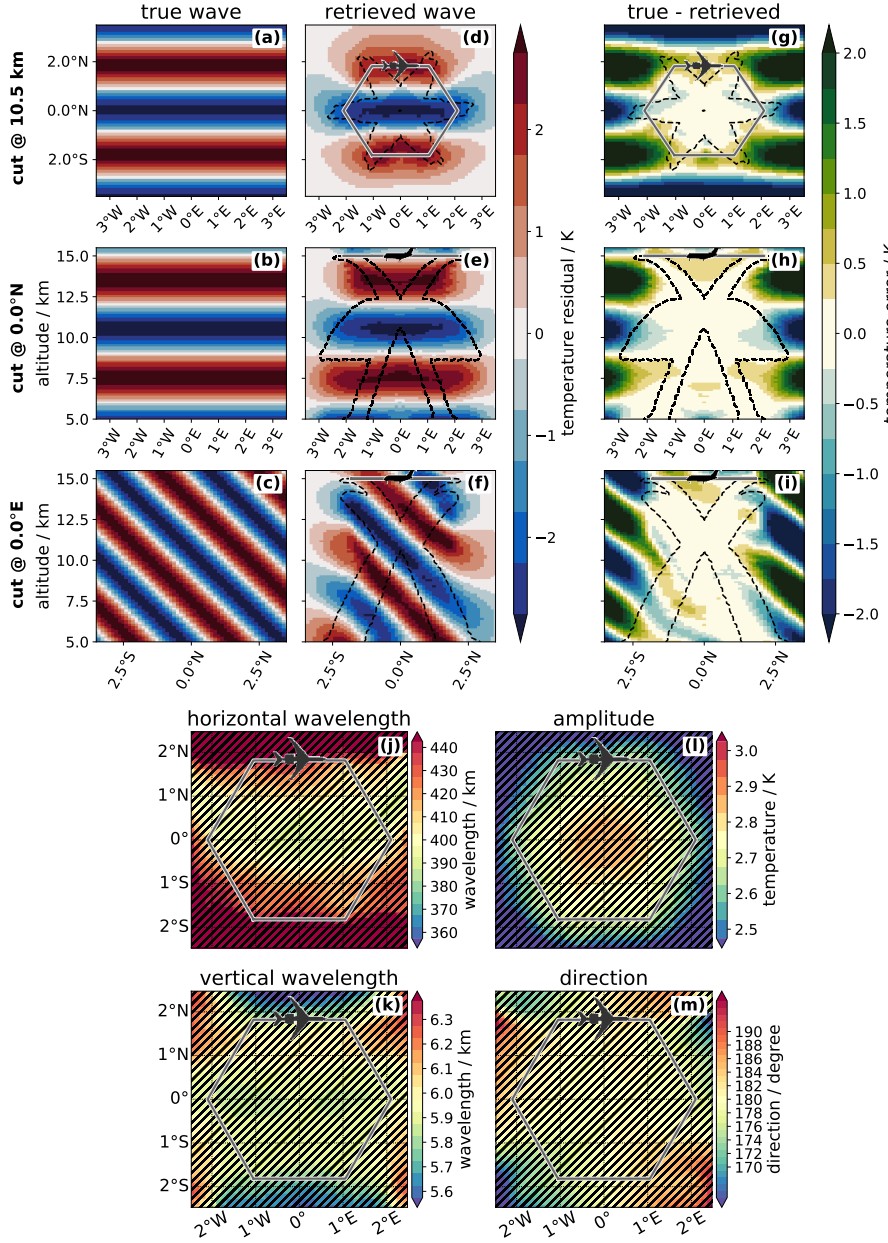

**Figure 3.** FAT retrieval and S3D results for a wave with 400 km horizontal, 6 km vertical wavelength, and horizontal wave direction $\varphi_t = 180°$. Panels **(a-c)** show the true wave, panels **(d-e)** the retrieved wave, and panels **(g-i)** the difference between both. The black dashed lines mark the area covered by tangent points. In panels **(j-m)** the S3D results horizontal wavelength $\lambda_h$, vertical wavelength $\lambda_z$, horizontal wave direction $\varphi$, and temperature amplitude $\hat{T}$ are pictured.





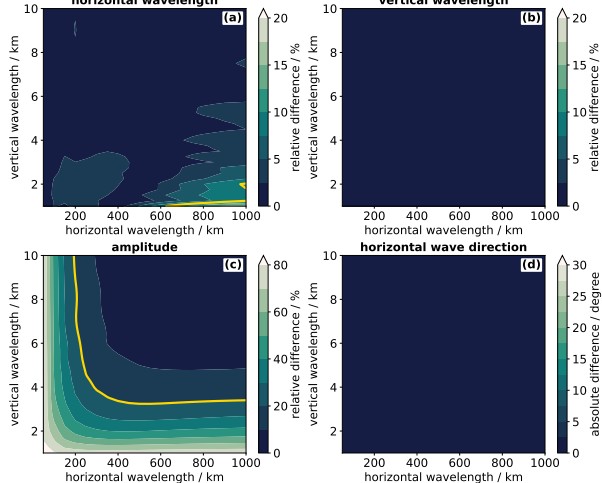

**Figure 4.** Specific observational filters for **(a)** horizontal wavelength $\lambda_h$, **(b)** vertical wavelength $\lambda_z$, **(c)** temperature amplitude $\hat{T}$, and **(d)** horizontal wave direction $\varphi$ for the FAT retrieval. The yellow lines mark errors of 10% for the horizontal wavelength (Panel **(a)**) and 20% for the amplitude (Panel **(c)**).

to an averaging over different wave phases. For the waves with short vertical wavelength the LOS crosses multiple opposite wave phases, which decreases the measurement signal. A similar dependence of the sensitivity on the alignment of phasefronts with LOS was observed for sub-limb viewers (Wu and Waters, 1996; McLandress et al., 2000).

All retrieved waves show a slight V shape pattern, which is more emphasized for the waves with short vertical wavelength.
This V shape is probably caused by the parabola shape of the LOS. The retrieval does not know, where along the line-of-sight how much of the measured radiation was emitted, unless crossing measurements give sufficient information. As the LAT has fewer measurements at different angles, the temperature signal is redistributed according to the weighting function (Fig. 2b) along the LOS. This can be nicely seen in the vertical cross sections in Fig. 5(g, o), where the warm temperature follows the LOS upwards behind the tangent point. This vertical shift of temperature also causes the northward oriented V shape in the
horizontal cross sections.

As already for the FAT case, the specific observational filters were calculated using the S3D fits of the LAT retrievals (Fig. 6). The deviations of horizontal and vertical wavelengths are mainly below 10%. Only for very short vertical and very long horizontal wavelengths errors of above 20% appear. This is probably due to the above mentioned V shape deformation of the wave, which is more difficult to fit with one single sinusoidal wave. The same problem appears for the horizontal wave
direction. For waves with short vertical and long horizontal wavelengths and, thus a strong V shape, the direction cannot be derived properly anymore. For the rest of the waves the direction error stays everywhere below 10°. The observational filter for the amplitude shows a similar pattern as for the FAT case.




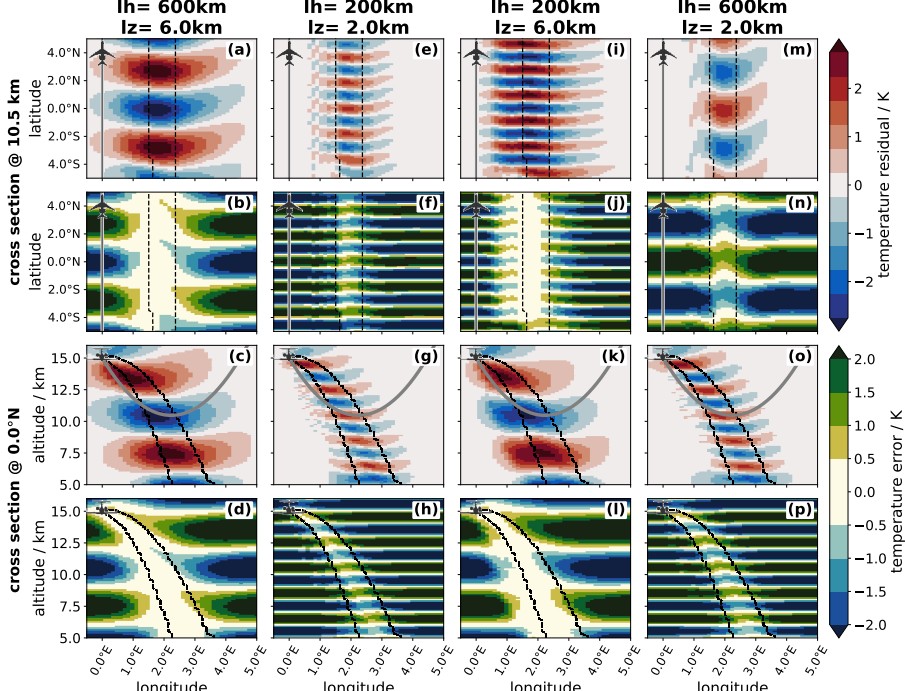

**Figure 5.** Cross sections of retrieved waves (first and third row) and differences between true and retrieved waves (second and forth row) of the LAT retrieval. The different columns show waves with different horizontal and vertical wavelengths. The true horizontal wave orientation of all waves is $\varphi_t = 180°$ and, thus, these waves have wavefronts perpendicular to the flight path. The black dashed lines mark the area covered by tangent points. The grey line in the vertical cross sections indicates a LOS for a measurement with $90°$ azimuth angle and tangent point altitude of $10.5\,\mathrm{km}$.

### 3.2.2 Dependence of retrieval results on the wave orientation

Due to the limited measurement sector, the orientation of the wave with respect to the instrument position might be important for LAT. Figure 7 depicts the retrieval results for waves with horizontal wave directions turned by $30°$ ($\varphi_t = 210°$) compared to those in Fig. 5. In the vertical, the wavefronts are tilted southward and, thus, towards the instrument. They decrease in height
5   with increasing distance to the flight path.

Overall, the structures are reproduced reasonably well. As for the perfectly perpendicularly-aligned waves already, waves with long vertical wavelengths (Fig. 7a–d and Fig. 7i–l) are reproduced better than waves with short vertical wavelengths (Fig. 7e–h and Fig. 7m–p).

Due to the tilt of the waves towards the aircraft, the LOS is partly aligned with the wavefronts before the tangent point.
10   This effect is stronger for steep waves such as in Fig. 7k than for relatively flat waves such as in Fig. 8c, g and o. Due to this alignment the area of best sensitivity is shifted towards the aircraft for the steep wave. Spreading the signal now around this





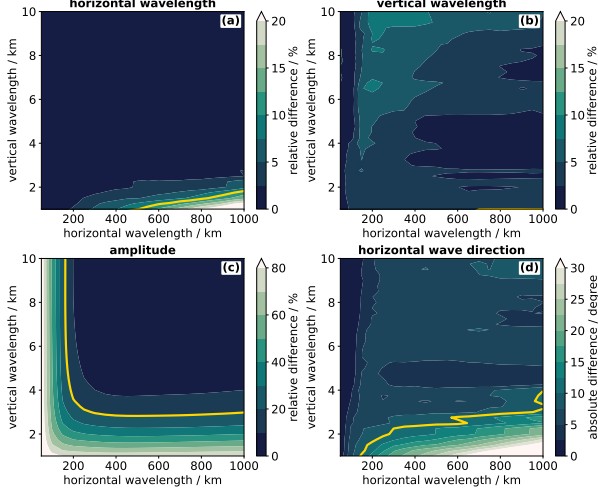

**Figure 6.** Specific observational filters for **(a)** horizontal wavelength $\lambda_h$, **(b)** vertical wavelength $\lambda_z$, **(c)** temperature amplitude $\hat{T}$, and **(d)** horizontal wave direction $\varphi$ for the LAT retrieval and true horizontal wave orientation $\varphi_t = 180°$ and, thus, these waves have wavefronts perpendicular to the flight path. The yellow lines mark errors of 10%, 10%, 20% and 10° in Panels **(a-d)**, respectively.

shifted sensitivity maximum, just spreads the signal along the same wave phase, as the LOS has little curvature in this region. Therefore, no strong shape deviation is observed. For the flat waves a similar V shape can be observed as for the waves in Fig. 5, due to a spreading of signal along LOS around the tangent point.

In the observational filter (Fig. 8) a small decrease in the quality of amplitude reproduction can be seen compared to the observational filter of perfectly east-west aligned waves (Fig. 6). However, the wavelengths and wave direction are barely influenced and reproduced at a similar high quality. The V shape of the waves only occurs outside the tangent point region, thus proper horizontal wave directions can be observed.

Fig. 9 shows the retrieval results for waves turned by -210° ($\varphi_t = 30°$) compared to Fig. 5. These waves are tilted northward and, thus, away from the flight path. Only for the wave with large horizontal and large vertical wavelength (Fig. 9a–d) the temperature amplitude is reproduced well within the tangent point region. However, the horizontal orientation in this area, which should be similar to Fig. 7a from north-west to south-east is not recovered. The same happens for waves with short vertical wavelengths (Fig. 9e–h and m–p): The information about the horizontal wave direction is lost within the retrieval. Again a V shape appears for all these waves. Due to the inverse vertical tilt compared to Fig. 7, the opening of the V shape is this time to the south.

For steep waves (Fig. 9k) the main signal is again shifted, this time behind the tangent point area, where the LOS and the wavefronts are well aligned. Thus the spreading of the signal does not influence these waves as strongly as the flat waves and the horizontal orientation does not get lost in the retrieval. The decreased amplitude compared to Fig. 7 can be explained by the fact that the maximum of the weighting function along LOS is located slightly before the tangent point (Fig. 2b).





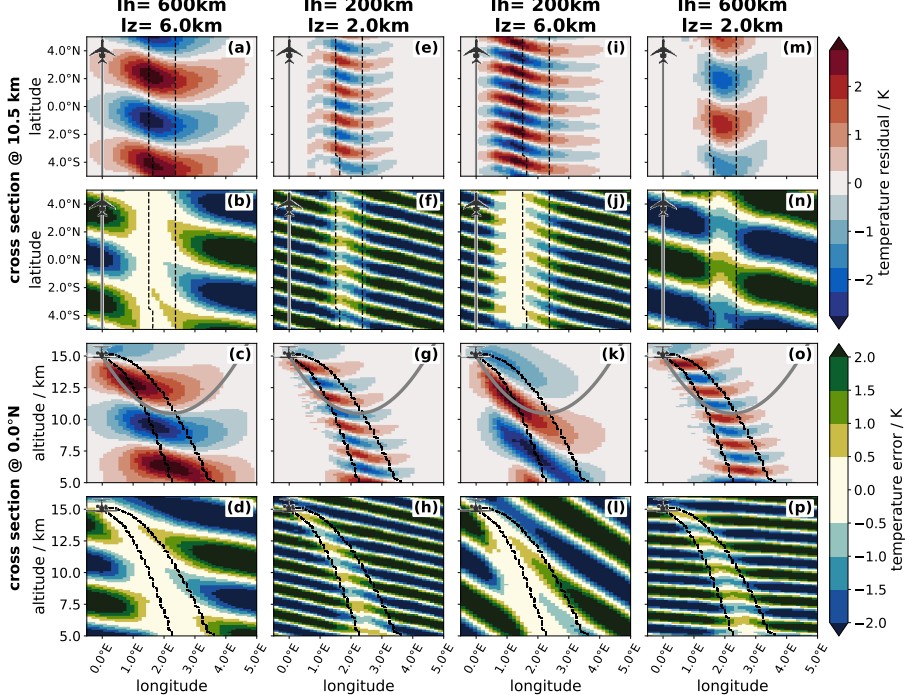

**Figure 7.** Cross sections of retrieved waves (first and third row) and differences between true and retrieved waves (second and forth row) of the LAT retrieval. The different columns show waves with different horizontal and vertical wavelengths. The true horizontal wave orientation of all waves is $\varphi_t = 210°$ and, thus, these waves are tilted towards the aircraft.

A similar picture is given from the observational filter in Fig. 10. Even though the amplitude is underestimated for very steep waves, the horizontal wave orientation can be derived accurately. However, the flatter the wave gets, the worse the derived horizontal wave direction. For waves with horizontal to vertical wavelengths ratio of above 200, the direction error exceeds 30°. Also the horizontal wavelengths reproduction is decreased somewhat compared to the two cases before (Fig. 6 and Fig. 8).

5      Further tests with horizontal wave direction $30° < \varphi_t < 90°$ and $210° < \varphi_t < 270°$ show a drastic decline in the amplitude sensitivity towards waves with short horizontal wavelengths. For waves tilted away from the flight path ($\varphi_t > 30°$) the fit quality of the horizontal wave direction and the horizontal wavelength decreases drastically already at $\varphi_t = 40°$.

These studies show that LAT applied to gravity waves gives best results for waves with wavefronts perpendicular to the flight path and, thus, horizontal wave vector $\varphi_t = 180°$. However, if the wave is slightly turned, the quality of the derived wave

10   parameters is not affected strongly as long as the wave is tilted towards the instrument ($180° <= \varphi_t <= 210°$). In general, waves are best retrieved when their aspect ratio of horizontal and vertical wavelengths, i.e. their steepness, is favourable for an alignment with the LOS. In these cases, tilts towards and away from the instrument may give reasonable results.





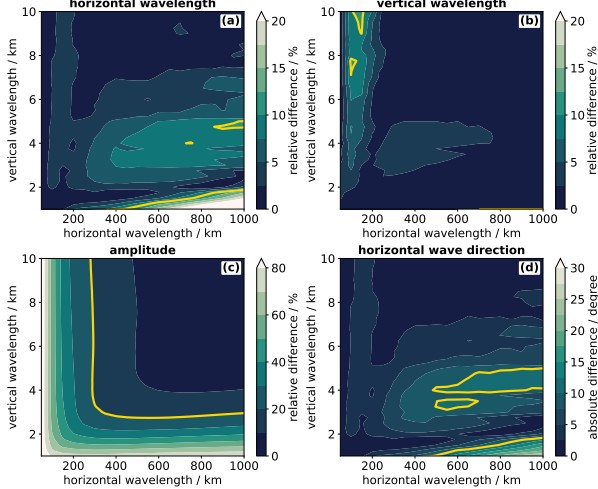

**Figure 8.** Specific observational filters for **(a)** horizontal wavelength $\lambda_h$, **(b)** vertical wavelength $\lambda_z$, **(c)** temperature amplitude $\hat{T}$, and **(d)** horizontal wave direction $\varphi$ for the LAT retrieval with true horizontal wave orientation $\varphi_t = 210°$ and, thus, waves, which are tilted towards the aircraft. The yellow lines mark errors of $10\%$, $10\%$, $20\%$ and $10°$ in Panels **(a-d)**, respectively.

### 3.3 Comparison of LAT and FAT results for a real measurement case on 25 January 2016 over Iceland

From December 2015 to March 2016, GLORIA was deployed on board of the German research aircraft HALO for a research campaign covering several scientific targets such as demonstrating the use of infrared limb imaging for gravity wave studies (GWEX), studying the full life cycle of a gravity wave (GW-LCYCLE), investigating the Seasonality of Air mass transport and

origin in the Lowermost Stratosphere (SALSA), and observing the Polar Stratosphere in a Changing Climate (POLSTRACC). On 25 January 2016, a research flight over Iceland investigated a GW excited at the Icelandic Mountains (Krisch et al., 2017). A linear flight leg of $500\,\mathrm{km}$ length crossing the wavefronts, was followed by a hexagonal flight pattern with $460\,\mathrm{km}$ diameter around the wave structure.

     Krisch et al. (2017) present the retrieval results of FAT using only measurements taken during the hexagonal flight. Fig. 11

compares these FAT results with LAT results using only measurements taken on the $500\,\mathrm{km}$ linear flight leg through the middle of the volume. In general the LAT results (Fig. 11 d-e) agree very well with the FAT results (Fig. 11 a-c) within the volume covered by both. FAT as well as the LAT retrieval, show a superposition of waves with longer and shorter horizontal wavelengths. Differences in strength and scale of the waves, for example in cross section #2, can be explained due to the different tangent point coverage of both methods. Especially higher altitudes in cross section #2 are not well covered with

tangent points in the FAT retrieval (Fig. 11c). This is probably the reason why the temperature residual is slightly lower in the FAT retrieval compared to LAT. Also the smaller scale waves in this region (Fig. 11f) are less prominent in the FAT retrieval (Fig. 11c).



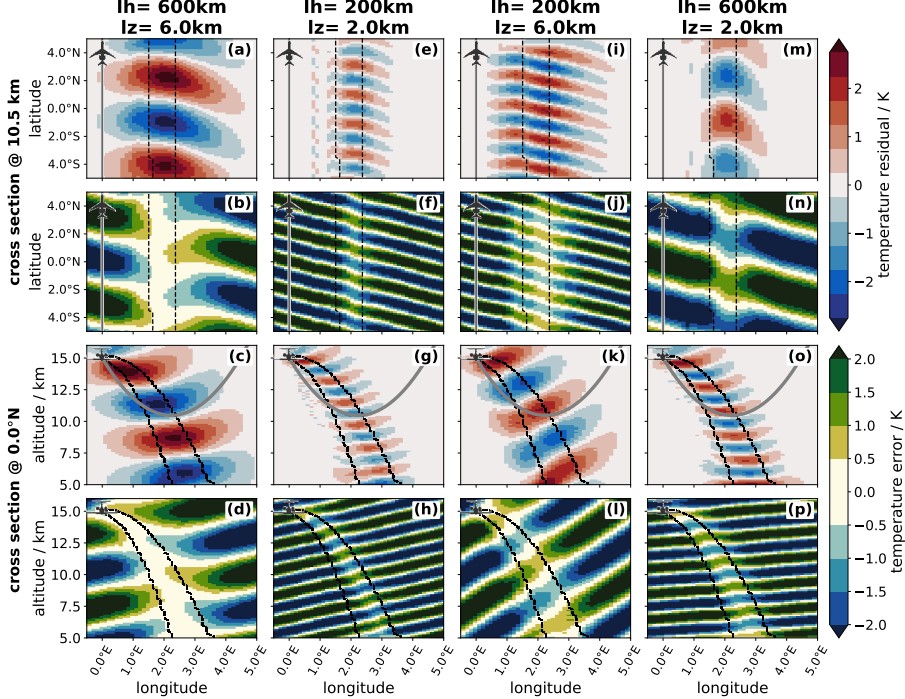

**Figure 9.** Cross sections of retrieved waves (first and third row) and differences between true and retrieved waves (second and forth row) of the LAT retrieval. The different columns show waves with different horizontal and vertical wavelengths. The true horizontal wave orientation of all waves is $\varphi_t = 30°$ and, thus, these waves are tilted away from aircraft.

A main advantage of 3-D tomographic measurement of GWs over conventional limb measurements is the ability to derive the horizontal wave direction. This is done by applying the S3D fitting routine. Figure 12 shows the wave parameters obtained from these fits for both cases. Within the confidence area of our fits, all wave parameters agree very well for both methods. The observed GW has a horizontal wavelength of around 200 km, a vertical wavelength around 5.5 km, amplitudes up to 2 K and a horizontal wave direction of 160°. Thus, the wave vector is turned by 20° compared to the flight direction. The wavefronts are tilted southward and, thus, away from the flight path. Figure 10 predicted for such waves a wavelength reproduction of more than 90% and an error in the estimation of the horizontal wave direction below 7.5°. This can be confirmed with the real measurement case.

## 4   Conclusions

This paper investigates the use of LAT applied to airborne limb imaging for gravity wave research. In contrast to FAT, which allows for the reconstruction of a large, cubic, 3-D volume, LAT can only reconstruct a band of 200 km around a banana-shaped vertical curtain parallel to the flight path. The horizontal resolution is 30 km in flight direction and 70 km perpendicular





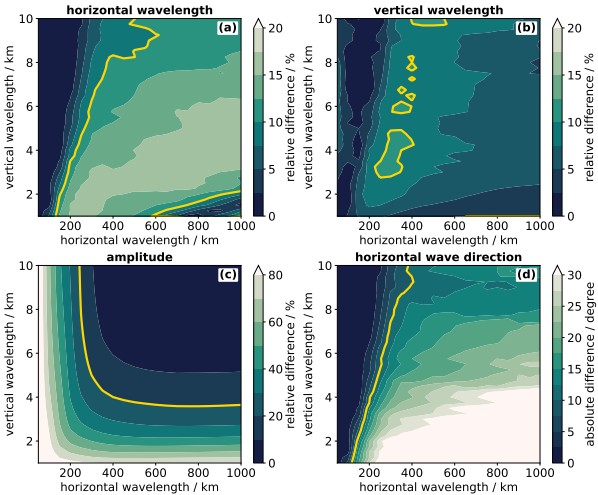

**Figure 10.** Specific observational filters for **(a)** horizontal wavelength $\lambda_h$, **(b)** vertical wavelength $\lambda_z$, **(c)** temperature amplitude $\hat{T}$, **(d)** and horizontal wave direction $\varphi$ for the LAT retrieval with true horizontal wave orientation $\varphi_t = 30°$ and, thus, waves, which are tilted away from aircraft. The yellow lines mark errors of 10%, 10%, 20% and 10° in Panels **(a-d)**, respectively.

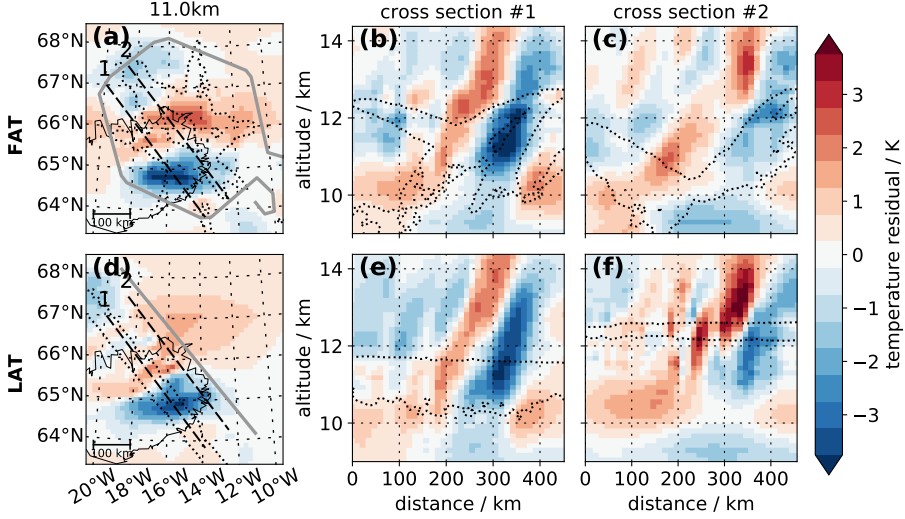

**Figure 11.** Comparison of FAT (a-c) and LAT (d-f) retrieval results for a research flight over Iceland on 25 January 2016. Shown are the temperature residuals after background removal as described in Sec. 2.5. The grey line indicates the part of the flight path, from which the measurements are used for the retrieval. The left column depicts horizontal crosssections at 11 km altitude, the two right columns present vertical cross sections along the dashed lines of the left column. The dotted lines mark the area covered by tangent points. Panels (a) to (c) are adapted from Krisch et al. (2017) Fig. 3.





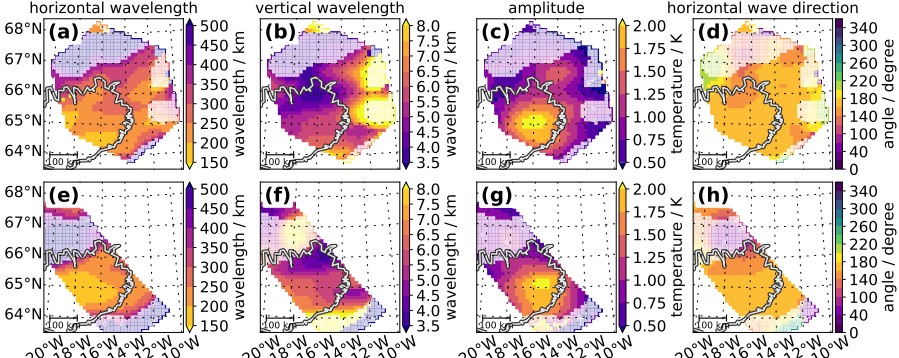

**Figure 12.** Wave parameters as obtained from the S3D fit with fitting cubes of 160 km x 160 km x 3.6 km at center height of 11.5 km for FAT (a-d) and LAT (e-h). Nonsignificant fitting results with wavelengths above 2.5 times the cube size are shaded. Panel (a) to (d) are adapted from Krisch et al. (2017) Fig. 5.

to flight direction. The vertical resolution is on the order of 400 m. This volume and resolution are sufficient to properly derive all important wave parameters such as the horizontal and vertical wavelengths, the amplitude, and the wave direction for waves with wavefronts perpendicular to the flight path. This is feasible due to the perfect alignment of wave phases and LOS and agrees well with earlier studies for other limb sounding concepts (Wu and Waters, 1996; McLandress et al., 2000; Ungermann

et al., 2010a).

The quality of the 3-D reconstruction strongly depends on the orientation of the wave with respect to the instrument. If the waves are slightly turned away from the perfect orientation the quality of the derived wave parameters is not strongly affected as long as the wavefront is tilted towards the instrument. If the wavefronts are tilted away from the instrument, the retrieval will create artefacts which reduce the quality of the derived horizontal wave directions and wavelengths. For waves with horizontal

wavelength under 300 km, the amplitude error is larger for waves with wavefronts tilted away from the instrument than for waves with wavefronts tilted towards the instrument. In general, the better the alignment of the wave phases and the LOS is, the more information is attained by the tomographic retrieval. Thus, steeper waves can be derived with better accuracy than flatter waves. For steep waves with a horizontal to vertical wavelength ratio below 200 correct wave directions can be derived independently of the tilt. However, for waves turned by more than 40° compared to the perfect, perpendicular case, the

reconstruction quality decreases drastically for all tested waves.

The capacity of LAT for GW research was demonstrated by comparing LAT and FAT for a real measurement case on 25 January 2016 above Iceland. The temperature residuals agree very well with each other. The wave parameters derived with a sinusoidal fitting routine yield similar results.

In summary, for many GW cases the observation in LAT mode can be recommended. However, for short scale waves FAT

is preferable due to the higher spatial resolution of 20 km x 20 km x 200 m. The slightly better accuracy of 0.5 K for FAT compared to 0.7 K for LAT also makes FAT favourable for low amplitude waves. Furthermore, when the precise orientation of





the wave cannot be predicted before the flight, FAT should be the method of choice. Nevertheless, for many other cases, LAT might be preferred due to its shorter acquisition time.

*Data availability.* The tomographic retrieval data used in Sec. 3.3 is available from the HALO database (Krisch, 2017, 2018).

*Competing interests.* The authors declare that they have no conflict of interest.

*Acknowledgements.* This work was partly supported by the Bundesministerium für Bildung und Forschung (BMBF) under project 01LG1206C (ROMIC/GW-LCYCLE), as well as by the European Space Agency (ESA) under contract 4000115111/15/NL/FF/ah (GWEX) and the Deutsche Forschungsgemeinschaft (DFG) project ER 474/4-2 (MS-GWaves/SV), which is part of the DFG researchers group FOR 1898 (MS-GWaves). The retrievals were performed on the JURECA supercomputer at the Jülich Supercomputing Center (JSC) as part of the JIEK72 project. We sincerely thank A. Dudhia, Oxford University, for providing the Reference Forward Model (RFM) used to calculate the
optical path and extinction cross-section tables required by our forward models. D. E. Kinnision, NCAR, is thanked for kindly providing the WACCM4 model data used in the retrieval. The European Centre for Medium-Range Weather Forecasts (ECMWF) is acknowledged for meteorological data support. The results are based on the efforts of all members of the GLORIA team, including the technology institutes ZEA-1 and ZEA-2 at Forschungszentrum Jülich and the Institute for Data Processing and Electronics at the Karlsruhe Institute of Technology. We would also like to thank the pilots and ground-support team at the Flight Experiments facility of the Deutsches Zentrum für Luft-
und Raumfahrt (DLR-FX).



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
