# Peer review of "Limited angle tomography of mesoscale gravity waves by the infrared limb-sounder GLORIA"

_Atmospheric Measurement Techniques, 2018_

## Referee Comment (RC1) · O. M. Christensen (Referee) · 27 Mar 2018

The paper shows the first results from the GLORIA instrument operating in the limited angle tomography mode. It highlight the benefits and drawbacks of this mode. It compares this mode to the full angle tomography mode, and perform a rigorous error analysis of the results. The paper is clearly written, interesting and easy to follow. I recommend publication with a few minor clarifications.

Minor comments P7l23: How is the accuracy and precision calculated? In particular what errors go into this error analysis? Are things like pointing error of the instrument, thermal noise, calibration error, instrument characterization errors, spectroscopic er-

rors or errors in the assumptions of the background atmosphere included? As short list is required for a reader to be able to judge whether all significant terms are included.

P8l12: please clarify what "25 and 60 neighboring points" means in terms of approximate distance or pressure.

P14l10: It is a bit unclear to me that "horizontal orientation in this area... is not recovered". For figure 9a, 9e, 9i, it does indeed look like the wave fronts are oriented from north-west to south-east. However, a degradation may be seen, but I have a bit hard time connecting the text with the figure. Perhaps explicitly writing out the directional error in the text (20 degrees?) would make it easier for the reader to follow.

P18 figure 11: In panel f) there is a large response in the retrieved values far below the tangent points. This is rather counter-intuitive, and the reason for it (I assume its caused by the horizontal regularization) should be written out in the text.

Editorial comments P2l24: "hpriozontal" -> horizontal P7l32: "ensures" is in my opinion too strong a word, unless the author can refer to some source which has done a rigorous error analysis on this method. Suggest that this sentence is rephrased.

---

## Referee Comment (RC2) · V. Hart (Referee) · 5 May 2018

This study provides a rigorous comparison between results from the GLORIA instrument, while operating in both full- and limited-angle tomography modes. It is well written and nicely organized. I believe the authors have done some excellent work and the study has significant potential. However, there are some critical details which should be addressed in more detail prior to publication. Specifically, I would like the authors to address the following.

Major Issues:

[Figure]

-P2_L23: You have described the FAT imaging volume as a cylinder, which is to be expected given the circular flight pattern described previously. You then go on to describe the LAT imaging region as simply a '3-D volume.' I suggest this imaging region, its shape, and dimensions require more of an explanation. In the conclusion (P17_L11), you refer to the FAT volume is 'cubic.' Is this a contradiction?

-The LAT flightpath geometry should be described in more detail. Specifically, the vertical curvature of the flight path is never discussed. Did this cause inconsistencies in image resolution as the altitude between GWs and GLORIA varied? Were edge effects or 'smearing' a problem as the altitude increased?

- You describe a 200-km horizontal extent perpendicular to the flight path (P2_L25). The resolution of overhead multi-angle images typically degrades moving towards the edge of the images (away from the nadir). Was this effect compensated for in any of your preprocessing steps?

-The initialization of tomography algorithms can have a profound effect on the resulting reconstructions and I believe your initialization (P6_L27) needs to be described in far more detail. This climatological field ac should be discussed further. How was it developed? How do you know it is accurate?

-It doesn't appear you have done any synthetic testing of your tomography algorithm. In this process, artificial projection (brightness) data are produced from a known synthetic structure and are then used to produce a 3D reconstruction. This result is compared with the original synthetic object to determine the accuracy of the algorithm. This approach can also be used to identify issues such as edge effects or biasing which results from the initialization model. Could you comment on this or perhaps include some sample synthetic results?

-P7_L21: You state the horizontal resolution along the flightpath is 30 km. Is this the image resolution? This isn't consistent with a field of view spanning only 100-200 km. -You might consider quantifying the spatial agreement between the resulting images in

Figure 11 (i.e., (b) and (e), (c) and (f)). Common metrics include the structural similarity index measurement (SSIM) or the Pearson correlation coefficient (PCC).

Minor Issues:

-P4_L24: 'with a 2-D detector array Friedl-Vallon et al. (2014).' This citation should be placed in brackets.

-P7_L21: Why did you decide to use values 'larger than half the maximum'?

-I am a little confused by your statement in the caption of Figure 2. You state the line-of-sight measurements assume a parabolic shape due to the cartesian coordinate system. Are you implying they are straight on a curved surface and then warped when plotted using straight axes?

-P10_L4: In discussing tomographic retrieval of gravity waves from atmospheric data, you have only cited one study. However, there have been multiple studies on this topic, using data from satellites such as ODIN and AIM. I suggest expanding these references.

-You need to be clearer about the differences between the rows in Figure 3. (a) and (c) show true waves which are nearly identical. Why are the retrieved waves so different?

-P19_L9: What kind of artifacts are you referring to? Can you describe them?

-It is unclear what 'lz' and 'lh' refer to in Figure 9.

---

## Author Comment (AC1) · 15 Jun 2018

Dear Dr. Christensen,

Thank you very much for these very helpful comments! We extended the explanation of the retrieval and the measurement diagnostics and improved the description of Figure 9 to make it easier for the reader to follow.

Please find detailed answers on all your comments below.

Sincerely, Isabell Krisch

[Figure]

**P7l23: How is the accuracy and precision calculated? In particular what errors go into this error analysis? Are things like pointing error of the instrument, thermal noise, calibration error, instrument characterization errors, spectroscopic errors or errors in the assumptions of the background atmosphere included? As short list is required for a reader to be able to judge whether all significant terms are included.**

A description of all errors implemented in the diagnostics is included in the manuscript. These errors include pointing errors, misrepresented background gases, uncertainties in spectral line characterization, calibration errors and measurement noise.

**P8l12: please clarify what "25 and 60 neighboring points" means in terms of approximate distance or pressure.**

Corresponds to 750km in the horizontal and 3km in the vertical. Has been included in the text.

**P14l10: It is a bit unclear to me that "horizontal orientation in this area. . . is not recovered". For figure 9a, 9e, 9i, it does indeed look like the wave fronts are oriented from north-west to south-east. However, a degradation may be seen, but I have a bit hard time connecting the text with the figure. Perhaps explicitly writing out the directional error in the text (20 degrees?) would make it easier for the reader to follow.**

The following text is added to better guide the reader through the Figure: Within the tangent point region (longitude between 1.5° and 2.5°) the horizontal wave fronts are oriented almost west to east. Behind the tangent point region (longitude above 2.5°), the wave fronts slowly approach the expected west-north-west to east-south-east orientation. The orientation error of this retrieval ranges between 30° and 10° depending on the area of interest.

**P18 figure 11: In panel f) there is a large response in the retrieved values far**

**below the tangent points. This is rather counter-intuitive, and the reason for it (I assume its caused by the horizontal regularization) should be written out in the text.**

This information has been included in the text, however at an earlier position (Section 3.1): Astonishingly, the retrieval can also reproduce some signal outside the area covered by tangent points. One reason for that is, the path of the LOS which goes through higher altitudes before and after the tangent point, thus collecting information there. Another reason are the horizontal and vertical correlation lengths of 10 km and 0.1 km, which are used for the retrieval and smear out the signal.

---

## Author Comment (AC2) · 15 Jun 2018

Dear Dr. Hart,

Thank you very much for these very helpful comments!

We included a better explanation of the measurement geometry to make it easier for the reader to follow the study. Also instrument effects are discussed in more detail. Further, we clarified that the paper is mainly based on synthetic data and only one Chapter uses real measurements.

Please find detailed answers on all your comments below.

[Figure]

Sincerely, Isabell Krisch

**P2L23: You have described the FAT imaging volume as a cylinder, which is to be expected given the circular flight pattern described previously. You then go on to describe the LAT imaging region as simply a '3-D volume.' I suggest this imaging region, its shape, and dimensions require more of an explanation. In the conclusion (P17L11), you refer to the FAT volume is 'cubic.' Is this a contradiction?**

A more detailed explanation of the volume reconstructed with LAT has been added to Section 2.2, as it would require too much space in the introduction (P2L23). Figure 2a has been changed to improve the understanding of the volume recovered with LAT. The term cubic in the conclusion has been replaced by "cylindrical" to avoid confusion.

**The LAT flightpath geometry should be described in more detail. Specifically, the vertical curvature of the flight path is never discussed. Did this cause inconsistencies in image resolution as the altitude between GWs and GLORIA varied? Were edge effects or 'smearing' a problem as the altitude increased?**

A more detailed explanation of the volume reconstructed with LAT has been added to Section 2.2. Figure 2a has been changed to improve the understanding of the volume recovered with LAT. The flight altitude has been kept constant during the whole study. This information is added to the text. This is also mostly the case during real aircraft measurements of gravity waves. Thus, smearing effects due to altitude changes do not play a role here.

**You describe a 200-km horizontal extent perpendicular to the flight path (P2L25). The resolution of overhead multi-angle images typically degrades moving towards the edge of the images (away from the nadir). Was this effect compensated for in any of your preprocessing steps?**

For the GLORIA retrieval presented in this paper, a region of interest of 48 x 128 pixels is used, albeit the optical system is designed for a region of interest of 128 x 128 pixels. In this configuration, the imaging quality is at, or close to, the diffraction limit for every pixel used and the edge effects are thus not a dominant error source in the retrieval process. The co-addition of pixels further reduces the relative impact of edge effects and the spectral errors are mostly corrected during level 1 processing (Friedl-Vallon et al., 2014; Kleinert et al., 2014). Furthermore, every single pixel has a point spread function (PSF) which is about 1.9 arcmin wide. As the tangent points for lower altitudes are further away from the airplane, the projected vertical PSF measured in meters is wider for lower tangent altitudes. This effect is taken into account in the radiative-transfer and instrument model of the retrieval (Ungermann et al., 2015).

**The initialization of tomography algorithms can have a profound effect on the resulting reconstructions and I believe your initialization (P6L27) needs to be described in far more detail. This climatological field ac should be discussed further. How was it developed? How do you know it is accurate?**

The retrieval setup was described in more detail. Especially the construction of the Covariance matrices was given in more detail. A detailed description of the used climatology and its accuracy is discussed in the given reference Remedios et al. (2007). The influence of the climatology on the result of the sensitivity study is negligible, because we are not interested in absolute temperature values but the perturbations. That is also one of the reasons, why the climatology is directly subtracted from the retrieval result before the analysis. Thus a detailed discussion of the climatology would exceed the scope of this paper.

**It doesn't appear you have done any synthetic testing of your tomography algorithm. In this process, artificial projection (brightness) data are produced from a known synthetic structure and are then used to produce a 3D reconstruction. This result is compared with the original synthetic object to determine the accuracy of the algorithm. This approach can also be used to identify issues such**

**as edge effects or biasing which results from the initialization model. Could you comment on this or perhaps include some sample synthetic results?**

The main point of this paper is to test the tomographic algorithm with synthetic data. To clarify this, the terminology has been changed and the before called "true" structures have been renamed to "synthetic". The authors hope that this helps to distinguish synthetic retrieval tests (Sections 3.1 & 3.2) from tests with real measurements (Section 3.3). The effects, which are introduced by the retrieval algorithm are discussed in section 3.2 extensively and include for example the "V-shape".

**P7L21: You state the horizontal resolution along the flightpath is 30 km. Is this the image resolution? This isn't consistent with a field of view spanning only 100-200 km.**

The horizontal field-of view of the detector is around 7km, which is mentioned in Section 2.2. The 100-200km are not a resolution but a coverage which is gained by overlapping forward and rearward looking measurements. We improved the description of the measurement geometry in Section 2.2 to avoid this confusion.

**You might consider quantifying the spatial agreement between the resulting images in Figure 11 (i.e., (b) and (e), (c) and (f)). Common metrics include the structural similarity index measurement (SSIM) or the Pearson correlation coefficient (PCC).**

The following has been included in the text: A more quantitative comparison of the similarities of both retrievals can be given by the Pearson correlation coefficient. Including only areas which are covered by tangent points gives a Pearson correlation coefficient of 0.91. Expanding this area to places with measurement content larger than 0.8 (includes areas crossed by a LOS before or after the tangent point) still leads to a Pearson correlation coefficient of 0.75. Thus, as expected the two retrievals are highly correlated.

**P7L21: Why did you decide to use values 'larger than half the maximum'?**

The resolution describes which area around a measurement point which influence its value. However, this influence can in theory reach till infinity. Atmospheric values closer to a measurement point in general have higher influence than values far away. One way to determine the influence of the atmosphere around the measurement point is to look at the averaging kernel matrix. Points with AVK smaller than the fwhm have lower impact and can therefore be neglected. This method is already used in 1-D and 2-D nadir and limb sounding (Rodgers, 2000; von Clarmann et al., 2009). In this paper we just extended this method to 3-D.

**I am a little confused by your statement in the caption of Figure 2. You state the line-of-sight measurements assume a parabolic shape due to the cartesian coordinate system. Are you implying they are straight on a curved surface and then warped when plotted using straight axes?**

Yes, we are implying that the originally straight lines are wrapped, when using straight axes in x, y and z. This has been clarified in the text.

**P10L4: In discussing tomographic retrieval of gravity waves from atmospheric data, you have only cited one study. However, there have been multiple studies on this topic, using data from satellites such as ODIN and AIM. I suggest expanding these references.**

Additional citations were added.

**You need to be clearer about the differences between the rows in Figure 3. (a) and (c) show true waves which are nearly identical. Why are the retrieved waves so different?**

Figure 3 shows only one wave, but multiple cross sections through the 3-D volume. The first row shows horizontal cross sections, rows 2 and 3 vertical cross sections along 0°N and 0°E, respectively. This has been clarified in the caption.

**P19L9: What kind of artifacts are you referring to? Can you describe them?**

The retrieval creates v-shaped phase fronts. This is added to the manuscript.

**It is unclear what 'lz' and 'lh' refer to in Figure 9.**

'lz' and 'lh' stand for the vertical and horizontal wavelengths of the waves, respectively. This has been clarified in the caption.

**References**

Friedl-Vallon, F., Gulde, T., Hase, F., Kleinert, A., Kulessa, T., Maucher, G., Neubert, T., Olschewski, F., Piesch, C., Preusse, P., Rongen, H., Sartorius, C., Schneider, H., Schönfeld, A., Tan, V., Bayer, N., Blank, J., Dapp, R., Ebersoldt, A., Fischer, H., Graf, F., Guggenmoser, T., Höpfner, M., Kaufmann, M., Kretschmer, E., Latzko, T., Nordmeyer, H., Oelhaf, H., Orphal, J., Riese, M., Schardt, G., Schillings, J., Sha, M. K., Suminska-Ebersoldt, O., and Ungermann, J.: Instrument concept of the imaging Fourier transform spectrometer GLORIA, Atmos. Meas. Tech., 7, 3565–3577, https://doi.org/10.5194/amt-7-3565-2014, 2014.

Kleinert, A., Friedl-Vallon, F., Guggenmoser, T., Höpfner, M., Neubert, T., Ribalda, R., Sha, M. K., Ungermann, J., Blank, J., Ebersoldt, A., Kretschmer, E., Latzko, T., Oelhaf, H., Olschewski, F., and Preusse, P.: Level 0 to 1 processing of the imaging Fourier transform spectrometer GLORIA: generation of radiometrically and spectrally calibrated spectra, Atmos. Meas. Tech., 7, 4167–4184, https://doi.org/10.5194/amt-7-4167-2014, 2014.

Remedios, J. J., Leigh, R. J., Waterfall, A. M., Moore, D. P., Sembhi, H., Parkes, I., Greenhough, J., Chipperfield, M., and Hauglustaine, D.: MIPAS reference atmospheres and comparisons to V4.61/V4.62 MIPAS level 2 geophysical data sets, Atmos. Chem. Phys. Discuss., 7, 9973–10 017, https://doi.org/10.5194/acpd-7-9973-2007, 2007.

Riese, M., Oelhaf, H., Preusse, P., Blank, J., Ern, M., Friedl-Vallon, F., Fischer, H., Guggenmoser, T., Hoepfner, M., Hoor, P., Kaufmann, M., Orphal, J., Ploeger, F., Spang, R., Suminska-Ebersoldt, O., Ungermann, J., Vogel, B., and Woiwode, W.: Gimballed Limb Observer for Radiance Imaging of the Atmosphere (GLORIA) scientific objectives, Atmos. Meas. Tech., 7, 1915–1928, https://doi.org/10.5194/amt-7-1915-2014, 2014.

Rodgers, C. D.: Inverse Methods for Atmospheric Sounding: Theory and Practice, vol. 2 of *Series on Atmospheric, Oceanic and Planetary Physics*, World Scientific, Singapore, 2000.

Ungermann, J., Blank, J., Dick, M., Ebersoldt, A., Friedl-Vallon, F., Giez, A., Guggenmoser, T., Höpfner, M., Jurkat, T., Kaufmann, M., Kaufmann, S., Kleinert, A., Krämer, M., Latzko, T., Oelhaf, H., Olchewski, F., Preusse, P., Rolf, C., Schillings, J., Suminska-Ebersoldt, O., Tan, V., Thomas, N., Voigt, C., Zahn, A., Zöger, M., and Riese, M.: Level 2 processing for the imaging Fourier transform spectrometer GLORIA: derivation and validation of temperature and trace gas volume mixing ratios from calibrated dynamics mode spectra, Atmos. Meas. Tech., 8, 2473–2489, https://doi.org/10.5194/amt-8-2473-2015, 2015.

von Clarmann, T., De Clercq, C., Ridolfi, M., Höpfner, M., and Lambert, J.-C.: The horizontal resolution of MIPAS, Atmos. Meas. Tech., 2, 47–54, https://doi.org/10.5194/amt-2-47-2009, 2009.

---

## Author Response (AR1)

**Point-by-point response for referee comments on "Limited angle tomography of mesoscale gravity waves by the infrared limb-sounder GLORIA"**

Isabell Krisch1, Jörn Ungermann1, Peter Preusse1, Erik Kretschmer2, and Martin Riese1

1Forschungszentrum Jülich, Institute of Energy- and Climate Research, Stratosphere (IEK-7), Jülich, Germany 2Karlsruhe Institute of Technology, Institute of Meteorology and Climate Research, Karlsruhe, Germany Correspondence: Isabell Krisch (i.krisch@fz-juelich.de)

**1 **Response to Referee Comment of Dr. Ole Christensen**

P7123: How is the accuracy and precision calculated? In particular what errors go into this error analysis? Are things like pointing error of the instrument, thermal noise, calibration error, instrument characterization errors, spectroscopic errors or errors in the assumptions of the background atmosphere included? As short list is required for a reader to be able to judge whether all significant terms are included.

5

A description of all errors implemented in the diagnostics is included in the manuscript. These errors include pointing errors, misrepresented background gases, uncertainties in spectral line characterization, calibration errors and measurement noise.

P8112: please clarify what "25 and 60 neighboring points" means in terms of approximate distance or pressure.

10 Corresponds to 750km in the horizontal and 3km in the vertical. Has been included in the text.

P14110: It is a bit unclear to me that "horizontal orientation in this area... is not recovered". For figure 9a, 9e, 9i, it does indeed look like the wave fronts are oriented from north-west to south-east. However, a degradation may be seen, but I have a bit hard time connecting the text with the figure. Perhaps explicitly writing out the directional error in the text (20 degrees?) would make it easier for the reader to follow.

The following text is added to better guide the reader through the Figure: Within the tangent point region (longitude between  $1.5^{\circ}$  and  $2.5^{\circ}$ ) the wave fronts are oriented almost west to east. Behind the tangent point region (longitude above  $2.5^{\circ}$ ), the wave fronts slowly approach the expected west-north-west to east-south-east orientation. The orientation error of this retrieval ranges between 30° and 10° depending on the area of interest.

15

P18 figure 11: In panel f) there is a large response in the retrieved values far below the tangent points. This is rather counterintuitive, and the reason for it (I assume its caused by the horizontal regularization) should be written out in the text.

20

This information has been included in the text, however at an earlier position (Section 3.1): The retrieval can also reproduce some signal outside the area covered by tangent points. One reason for that is, the path of the LOS which goes through higher altitudes before and after the tangent point, thus collecting information there. Another reason are the horizontal and vertical correlation lengths of 100 km and 1 km, which are used for the retrieval and smear out the signal.

**5 2 Response to Referee Comment of Dr. Vern Hart**

20

P2L23: You have described the FAT imaging volume as a cylinder, which is to be expected given the circular flight pattern described previously. You then go on to describe the LAT imaging region as simply a '3-D volume.' I suggest this imaging region, its shape, and dimensions require more of an explanation. In the conclusion (P17L11), you refer to the FAT volume is 'cubic.' Is this a contradiction?

10 A more detailed explanation of the volume reconstructed with LAT has been added to Section 2.2, as it would require too much space in the introduction (P2L23). Figure 2a has been changed to improve the understanding of the volume recovered with LAT. The term cubic in the conclusion has been replaced by "cylindrical" to avoid confusion.

The LAT flightpath geometry should be described in more detail. Specifically, the vertical curvature of the flight path is 15 never discussed. Did this cause inconsistencies in image resolution as the altitude between GWs and GLORIA varied? Were edge effects or 'smearing' a problem as the altitude increased?

A more detailed explanation of the volume reconstructed with LAT has been added to Section 2.2. Figure 2a has been changed to improve the understanding of the volume recovered with LAT. The flight altitude has been kept constant during the whole study. This information is added to the text. This is also mostly the case during real aircraft measurements of gravity waves. Thus, smearing effects due to altitude changes do not play a role here.

You describe a 200-km horizontal extent perpendicular to the flight path (P2L25). The resolution of overhead multi-angle images typically degrades moving towards the edge of the images (away from the nadir). Was this effect compensated for in any of your preprocessing steps?

- For the GLORIA retrieval presented in this paper, a region of interest of 48 x 128 pixels is used, albeit the optical system is designed for a region of interest of 128 x 128 pixels. In this configuration, the imaging quality is at, or close to, the diffraction limit for every pixel used and the edge effects are thus not a dominant error source in the retrieval process. The co-addition of pixels further reduces the relative impact of edge effects and the spectral errors are mostly corrected during level 1 processing (Friedl-Vallon et al., 2014; Kleinert et al., 2014). Furthermore, every single pixel has a point spread function (PSF) which is
- 30 about 1.9 arcmin wide. As the tangent points for lower altitudes are further away from the airplane, the projected vertical PSF measured in meters is wider for lower tangent altitudes. This effect is taken into account in the radiative-transfer and instrument model of the retrieval (Ungermann et al., 2015).

The initialization of tomography algorithms can have a profound effect on the resulting reconstructions and I believe your initialization (P6L27) needs to be described in far more detail. This climatological field ac should be discussed further. How was it developed? How do you know it is accurate?

The retrieval setup was described in more detail. Especially the construction of the Covariance matrices was given in more 5 detail.

A detailed description of the used climatology and its accuracy is discussed in the given reference Remedios et al. (2007). The influence of the climatology on the result of the sensitivity study is negligible, because we are not interested in absolute temperature values but the perturbations. That is also one of the reasons, why the climatology is directly subtracted from the retrieval result before the analysis. Thus a detailed discussion of the climatology would exceed the scope of this paper.

10

It doesn't appear you have done any synthetic testing of your tomography algorithm. In this process, artificial projection (brightness) data are produced from a known synthetic structure and are then used to produce a 3D reconstruction. This result is compared with the original synthetic object to determine the accuracy of the algorithm. This approach can also be used to identify issues such as edge effects or biasing which results from the initialization model. Could you comment on this or perhaps include some sample synthetic results?

15 perhaps include some sample synthetic results?

The main point of this paper is to test the tomographic algorithm with synthetic data. To clarify this, the terminology has been changed and the before called "true" structures have been renamed to "synthetic". The authors hope that this helps to distinguish synthetic retrieval tests (Sections 3.1 & 3.2) from tests with real measurements (Section 3.3). The effects, which are introduced by the retrieval algorithm are discussed in section 3.2 extensively and include for example the "V-shape".

20

P7L21: You state the horizontal resolution along the flightpath is 30 km. Is this the image resolution? This isn't consistent with a field of view spanning only 100-200 km.

The horizontal field-of view of the detector is around 7km, which is mentioned in Section 2.2. The 100-200km are not a resolution but a coverage which is gained by overlapping forward and rearward looking measurements. We improved the description of the measurement geometry in Section 2.2 to avoid this confusion.

25

You might consider quantifying the spatial agreement between the resulting images in Figure 11 (i.e., (b) and (e), (c) and (f)). Common metrics include the structural similarity index measurement (SSIM) or the Pearson correlation coefficient (PCC). The following has been included in the text: A more quantitative comparison of the similarities of both retrievals can be given

- 30 by the Pearson correlation coefficient. Including only areas which are covered by tangent points gives a Pearson correlation coefficient of 0.91. Expanding this area to places with measurement content larger than 0.8 (includes areas crossed by a LOS before or after the tangent point) still leads to a Pearson correlation coefficient of 0.75. Thus, as expected the two retrievals are highly correlated.
- 35 P7L21: Why did you decide to use values 'larger than half the maximum'?

The resolution describes which area around a measurement point which influence its value. However, this influence can in theory reach till infinity. Atmospheric values closer to a measurement point in general have higher influence than values far away. One way to determine the influence of the atmosphere around the measurement point is to look at the averaging kernel matrix. Points with AVK smaller than the fwhm have lower impact and can therefore be neglected. This method is already

5 used in 1-D and 2-D nadir and limb sounding (Rodgers, 2000; von Clarmann et al., 2009). In this paper we just extended this method to 3-D.

I am a little confused by your statement in the caption of Figure 2. You state the line-of-sight measurements assume a parabolic shape due to the cartesian coordinate system. Are you implying they are straight on a curved surface and then warped

10 when plotted using straight axes?

Yes, we are implying that the originally straight lines are wrapped, when using straight axes in x, y and z. This has been clarified in the text.

P10L4: In discussing tomographic retrieval of gravity waves from atmospheric data, you have only cited one study. However,
there have been multiple studies on this topic, using data from satellites such as ODIN and AIM. I suggest expanding these references.

Additional citations were added.

You need to be clearer about the differences between the rows in Figure 3. (a) and (c) show true waves which are nearly identical. Why are the retrieved waves so different?

Figure 3 shows only one wave, but multiple cross sections through the 3-D volume. The first row shows horizontal cross sections, rows 2 and 3 vertical cross sections along  $0^{\circ}$ N and  $0^{\circ}$ E, respectively. This has been clarified in the caption.

P19L9: What kind of artifacts are you referring to? Can you describe them?

25 The retrieval creates v-shaped phase fronts. This is added to the manuscript.

It is unclear what 'lz' and 'lh' refer to in Figure 9.

'lz' and 'lh' stand for the vertical and horizontal wavelengths of the waves, respectively. This has been clarified in the caption.

30
$$J(\boldsymbol{a}) = \left(\mathbf{F}(\boldsymbol{a}) - \boldsymbol{y}\right)^T \mathbf{S}_{\epsilon}^{-1} \left(\mathbf{F}(\boldsymbol{a}) - \boldsymbol{y}\right) + \left(\boldsymbol{a} - \boldsymbol{a}_a\right)^T \mathbf{S}_a^{-1} \left(\boldsymbol{a} - \boldsymbol{a}_a\right) \to \min$$
(3)

**Table 1.** Standard deviations, vertical  $(c_z)$  and horizontal  $(c_h)$  correlation lengths used for the tomographic retrieval.

| atmospheric quantity              | $\stackrel{\sigma}{\sim}$         | $\underbrace{w_0}$                    | $\underline{w}_{1}$               | cz                                             | c p                            |
|-----------------------------------|-----------------------------------|---------------------------------------|-----------------------------------|------------------------------------------------|---------------------------------------|
| temperature                       | 1 K                        | $\underbrace{10^{-3}}_{\sim\sim}$     | $\underbrace{10^{-2}}_{\sim\sim}$ | 1.0 km                                  | 100 km                         |
| Q3                                | $\underbrace{141}{}\mathrm{ppbV}$ | $\underbrace{10^{-6}}_{\sim\sim}$     | $\underbrace{10^{-5}}_{\sim}$     | $\underline{\overset{8.0}{\times}}\mathrm{km}$ | $\underbrace{6400}_{6400}\mathrm{km}$ |
| $\underline{CCl}_{\underline{4}}$ | $\underbrace{13}{\text{pptV}}$    | $\underbrace{10^{-5}}_{\sim\sim}$     | $\underbrace{10^{-4}}_{\sim\sim}$ | $\underline{2.0}\mathrm{km}$                   | $\underline{800}\mathrm{km}$          |
| HNO3                              | $\underline{987}\mathrm{pptV}$    | $\underbrace{10^{-7}}_{\sim\sim\sim}$ | $\underbrace{10^{-4}}_{\sim\sim}$ | $3.2 \mathrm{km}$                              | $\underbrace{1280}_{}\mathrm{km}$     |

with  $\mathbf{S}_{\epsilon} \in \mathbb{R}^{o \times o}$  the measurement error covariance matrix and  $\mathbf{S}_{a} \in \mathbb{R}^{n \times n}$  the covariance matrix of the atmospheric state vector. As an apriori state  $a_{a}$  the climatological field  $a_{c}$  is used.

The first term of the costfunction represents the inversion of the forward model, which can have many different mathematical solutions. To choose a physically meaningful solution, a regularisation term using a covariance matrix  $S_a$  is added. This covariance matrix is constructed as follows:

$$\mathbf{S}_{a}^{-1} = \frac{w_{0}}{\sigma^{2}} ||\boldsymbol{a}||^{2} + \frac{w_{1}}{\sigma^{2}} \left( ||c_{z}\frac{\partial}{\partial z}\boldsymbol{a}||^{2} + ||c_{h}\frac{\partial}{\partial x}\boldsymbol{a}||^{2} + ||c_{h}\frac{\partial}{\partial y}\boldsymbol{a}||^{2} \right).$$

$$(4)$$

The standard deviations  $\sigma$ , weighting factors  $w_0$  and  $w_1$ , and the correlations lengths  $c_z$  and  $c_h$  used for the retrieval are given in Tab. 1. These factors are chosen ad-hoc and cannot be interpreted directly as physically meaningful correlation lengths. A more physical regularisation scheme is currently under development and will be described by Krasauskas et al. (2018).

10 This minimization problem is solved with a truncated conjugate gradient based trust region scheme. More details on the retrieval algorithms used for GLORIA Level 2 processing are described by Ungermann et al. (2015).

5

Since the temperature perturbations due to the wave are small compared to the background temperature  $T_c$ , the retrieval can be linearized around this background temperature (Rodgers, 2000; Ungermann et al., 2010a):

$$\boldsymbol{y} - \boldsymbol{y}_a = \mathbf{F}'(\boldsymbol{a}_a)(\boldsymbol{a} - \boldsymbol{a}_a) + \boldsymbol{\epsilon}.$$
(5)

15  $\mathbf{F}'(a_a) = \frac{\partial \mathbf{F}}{\partial a}\Big|_{a_a}$  is the Jacobian matrix of the forward model evaluated at  $a_a$  and  $y_a = F(a_a)$  are the simulated radiances of the background state. With the retrieval gain matrix  $\mathbf{G}(a_a) = (\mathbf{F}'(a_a)^T \mathbf{S}_{\epsilon}^{-1} \mathbf{F}'(a_a) + \mathbf{S}_{a}^{-1})^{-1} \mathbf{F}'(a_a)^T \mathbf{S}_{\epsilon}^{-1}$  and the Jacobian matrix  $\mathbf{F}'(a_a)$  the averaging kernel matrix  $\mathbf{A}(a_a) = \mathbf{G}(a_a)\mathbf{F}'(a_a)$  can be calculated, which converts the true temperature perturbation  $w_t = a_t - a_a$  synthetic temperature perturbation  $w_s = a_s - a_a$  into the retrieved temperature perturbation  $w_r = a_r - a_a$ :

20
$$\mathbf{G}(\boldsymbol{a}_a)(\boldsymbol{y}-\boldsymbol{y}_a) = \mathbf{G}(\boldsymbol{a}_a)(\mathbf{F}'(\boldsymbol{a}_a)(\boldsymbol{a}-\boldsymbol{a}_a)+\boldsymbol{\epsilon})$$
 (6)

$$\boldsymbol{a}_r - \boldsymbol{a}_a = \mathbf{A}(\boldsymbol{a}_a)(\boldsymbol{a}_{\underline{t}\,\underline{s}} - \boldsymbol{a}_a) + \mathbf{G}(\boldsymbol{a}_a)\boldsymbol{\epsilon} \tag{7}$$

$$\boldsymbol{w}_r = \mathbf{A}(\boldsymbol{a}_a)\boldsymbol{w}_{ts} + \mathbf{G}(\boldsymbol{a}_a)\boldsymbol{\epsilon}.$$
(8)

**Table 2. Systematic error sources included in the retrieval diagnostics with respective standard deviations and correlation lengths.**

| Error source                                       | standard deviation                         | correlation lengths                                           |
|----------------------------------------------------|--------------------------------------------|---------------------------------------------------------------|
| pointing errors
misrepresented background gases | 0.05°
taken from Remedios et al. (2007) | vertical: ∞, temporal: 0
taken from Remedios et al. (2007) |
| uncertainties in spectral line characterization    | _5%                                        | temporal: ∞                                                   |
| calibration errors gain                            | $\frac{1\%}{5}$ nW                         | temporal: $\infty$                                            |
| canoration errors offset                           | 2.11.00                                    | temporal. O                                                   |

[revised manuscript text omitted]

This relative error is a helpful measure for the reproducibility of GWs by the measurement setup and the retrieval concept. However, it does not give detailed information upon which wave parameters can be derived. Thus, we further define more specific relative errors for the important gravity wave parameters, horizontal  $\lambda_h$  and vertical  $\lambda_z$  wavelengths, amplitude  $\hat{T}$  and horizontal wave orientation  $\varphi$ :

$$S_{\xi} = \sum_{\boldsymbol{x}_i \in A} \underbrace{\frac{\xi_t(\boldsymbol{x}_i) - \xi_r(\boldsymbol{x}_i)}{\xi_t(\boldsymbol{x}_i)}}_{\underset{\boldsymbol{x}_i \in \mathbf{X}_i}{\underbrace{\xi_s(\boldsymbol{x}_i)}}} \underbrace{\xi_s(\boldsymbol{x}_i) - \xi_r(\boldsymbol{x}_i)}_{\underset{\boldsymbol{x}_i \in \mathbf{X}_i}{\underbrace{\xi_s(\boldsymbol{x}_i)}}} \text{ for } \xi \in \lambda_h, \lambda_z, \hat{T}, \varphi$$
(12)

These more specific relative errors help to define the quality of our measurement for gravity wave analysis.

The observational filter O = 1 - S is a measure of the sensitivity of an instrument and defines which gravity waves can be detected. The knowledge of the observational filter is necessary for meaningful comparisons of measurements from different instruments or measurement and model results (Alexander, 1998; Preusse et al., 2002; Ern et al., 2006; Ern and Preusse, 2012; Trinh et al., 2016).

**3 Results and discussion**

**10 3.1 Full angle tomography (FAT)**

Former studies demonstrated the feasibility of 2-D tomography of rearward looking satellite instruments for the retrieval of 3-D atmospheric structures such as gravity waves (Degenstein et al., 2004; Ungermann et al., 2010a; Hultgren et al., 2013). However, the concept of 3-D tomography with sidewards looking airborne instruments (Riese et al., 2014) or sub-limb instruments (Song et al., 2017; Hart et al., 2018) is quite different and, thus, may exhibit different characteristics. In 2-D tomography a

15 volume is reconstructed from rearward looking measurements on a moving platform which slice the volume into multiple 2-D images. In 3-D FAT, a volume is reconstructed from measurements at different sides all around the volume. Thus, with FAT the problem of wave orientation with respect to instrument and flight path (Ungermann et al., 2010a) does not appear.

Figure 3 shows a 3-D FAT of a GW with horizontal wavelength of 400 km, vertical wavelength of 6 km and horizontal wave direction  $\varphi_t = 180^\circ \varphi_s = 180^\circ$ . Pictured are three cross sections through the 3-D volume: at 10.5 km altitude (first row), at

20 0° N (second row), and at 0° E (third row). The first column shows the true synthetic wave, the second column the retrieved wave and the third column the difference of both. This synthetic wave has east-west oriented phase fronts (Fig. 3a) and is tilted to the south (Fig. 3c).

In general, the signal is well reproduced by the retrieval. Within the tangent point area (dotted lines) the temperature error is below 0.5 K. This is in good agreement with the determined accuracy of 0.5 K (Sec. 2.4). The retrieval can also reproduce

25 some signal outside the area covered by tangent points. One reason is, the path of the LOS which goes through higher altitudes before and after the tangent point, thus collecting information there. Another reason are the horizontal and vertical correlation lengths of 100 km and 1 km, which are used for the retrieval and smear out the signal.

A S3D fit (Sec. 2.6) was performed for this retrieval at 10.5 km altitude with a cube size of 5 km x 400 km x 400 km. The results of this fit can be seen in Fig. 3j-m. Within the hexagonal flight pattern the horizontal and vertical wavelength, and the

30 horizontal wave direction are well reproduced. The original amplitude of 3 K is underestimated by 0.1 K.